# A dynamic eco-evolutionary model predicts slow response of alpine plants to climate warming

Olivier Cotto[1,2], Johannes Wessely[3], Damien Georges[4], Günther Klonner[3], Max Schmid[1], Stefan Dullinger[3], Wilfried Thuiller[4] & Frédéric Guillaume[1]

Withstanding extinction while facing rapid climate change depends on a species' ability to track its ecological niche or to evolve a new one. Current methods that predict climate-driven species' range shifts use ecological modelling without eco-evolutionary dynamics. Here we present an eco-evolutionary forecasting framework that combines niche modelling with individual-based demographic and genetic simulations. Applying our approach to four endemic perennial plant species of the Austrian Alps, we show that accounting for eco-evolutionary dynamics when predicting species' responses to climate change is crucial. Perennial species persist in unsuitable habitats longer than predicted by niche modelling, causing delayed range losses; however, their evolutionary responses are constrained because long-lived adults produce increasingly maladapted offspring. Decreasing population size due to maladaptation occurs faster than the contraction of the species range, especially for the most abundant species. Monitoring of species' local abundance rather than their range may likely better inform on species' extinction risks under climate change.

[1] Department of Evolutionary Biology and Environmental Studies, University of Zurich, Winterthurerstrasse 190, Zurich CH-8057, Switzerland. [2] CEFE-CNRS, 1919 Route de Mende, Montpellier 5 34293, France. [3] Department of Botany and Biodiversity Research, Faculty of Life Sciences, University of Vienna, Rennweg 14, Vienna 1030, Austria. [4] Univesity Grenoble Alpes, CNRS, Laboratoire d'Écologie Alpine, Grenoble F-38000, France. Correspondence and requests for materials should be addressed to O.C. (email: oliviercotto@hotmail.fr) or to F.G. (email: frederic.guillaume@ieu.uzh.ch).

Following reports of the current detrimental impact of climate change on biodiversity[1], there is an urgent need to understand and predict the consequences of climate change on species persistence to inform conservation planning and provide guidelines and tools for climate change adaptation and mitigation[2]. Methods predicting decadal or centennial biodiversity losses over large geographical ranges use species distribution models (SDMs) to forecast the loss of suitable habitats caused by climate changes in different species and associate losses of species' range size with extinction risks[3,4]. However, SDM predictions have large uncertainties because of the correlative approach implemented in SDMs[5] and because they do not account for eco-evolutionary processes[6]. In particular, SDM projections assume niche conservatism; the described eco-climatic 'envelopes' do not evolve even though populations may adapt to temporal shifts of their local conditions. With the current pace of climate change, species are likely to undergo strong selective pressures[7–9] that may surpass their ability to adapt to the changing environment[10,11]. Yet, there is mounting evidence for the occurrence of evolutionary responses to ongoing climate change in several species[12–15]. New methods are thus needed that account not only for possible evolutionary responses (for example, ref. 16) but also for the feedback between adaptive and demographic processes to better predict species responses to climate and global changes[2,6,17,18]. Our aim here is to show how such a method can be implemented and applied to better understand species' range shifts under climate change in Alpine habitats.

Mountainous environments harbour a high rate of endemism and are considered particularly vulnerable to climate warming as they combine a steep climatic gradient with a decline of available area with altitude, especially in 'pyramid-shaped' mountain ranges like the European Alps[19]. Attempts to quantify how climate change would affect mountain biodiversity have either relied on spatial projections of species' climatic niches[20,21], a combination of niche-based and demographic modelling[22] or on spatially explicit dynamic vegetation models[23]. None of these studies has considered that evolution can co-determine and modify the response of mountain species to climate warming. Theoretical models, however, have demonstrated the key roles of migration and adaptation in the response of species to environmental gradients both in time (for example, ref. 24) and space (for example, refs 25,26). In addition, some experimental studies have shown that rapid evolution can rescue microbial species from local extinction under environmental change[27], although others have suggested that plant or insect species may lack sufficient genetic variation to adapt to rapid environmental changes[10,28,29]. It thus remains unclear how local additive genetic variance, life history, landscape structure, and dispersal interactively constrain, or promote, rapid evolutionary adaptation of species to a changing environment[8,17,30]. To add to this uncertainty, only a handful of studies have incorporated evolutionary processes into biodiversity models and most of them concern species with short generation times and fast growth (for example, dengue mosquitoes[31] or flies[16]). Theoretical reasoning suggests that longer-lived species in stressful environments like high mountains should display slower evolutionary responses[29,32], but no study has yet evaluated how adaptive evolution helps such species cope with climate change.

Here we fill this gap by assessing the role of evolution in the response of long-lived mountain plant species to climate change. To do so, we developed a new method that combines niche-based projections from SDMs and empirical data to parameterize individual-based, genetically and spatially explicit, stochastic simulations (that is, dynamic eco-evolutionary models, DEEMs). DEEMs assume that local populations on a landscape (for example, grid cells) adapt to their local environmental conditions through genetic evolution, while explicitly accounting for the stochastic processes of individual birth, death and migration in an age-structured demographic model. As a first step, we use static ecological niche models (SENMs) based on SDMs (see Methods) to predict the current distribution of a species in a study area as a function of spatial variation in environmental conditions. This predicted distribution pattern is then used to initialize populations in DEEMs, which subsequently simulate changes in the distribution and adaptation of plant individuals as driven by scenarios of climatic (or other environmental) change. We illustrate the approach by applying it to four species selected to be as different as possible in a Hill-Smith analysis (see Methods and Supplementary Fig. 1) performed on a list of 24 endemic Alpine plants previously modelled by Dullinger et al.[22]. The selected species, Campanula pulla L., Primula clusiana L., Festuca pseudodura Steud, and Dianthus alpinus L., differ with respect to their life histories (Supplementary Fig. 1), and ecological traits (niche width, Supplementary Table 1). The study area comprises 15 landscapes in the Austrian Alps where the four endemic species occur (Methods, Supplementary Figs 2 and 3). The initial species distribution was modelled as a function of three environmental variables: bedrock carbonates, mean annual temperature, and mean annual precipitation. Temporal changes in the mean annual temperature and precipitation were driven by three climatic forecasts of the Intergovernmental Panel on Climate Change (see Supplementary Methods) from 2010 to 2090. We further extend the DEEM simulations by a period of constant climate between 2090 and 2150 to evaluate for a mid-term impact of species maladaptation on population persistence, called an extinction debt[22]. We then compare range projections from the DEEM to SENM projections for the same climatic series to investigate how demographic and evolutionary processes may modify the range dynamics of the species under a warming climate. Finally, we perform a sensitivity analysis of the DEEM projections to parameters for which no precise estimates were available: additive genetic variance for the traits (via mutation rates), the strength of selection on seedling survival and the adult survival rate.

We find that evolutionary adaptation is unlikely to prevent the decrease in species range predicted under the selected climate change scenarios. The long adult lifespan of the studied species favours population persistence but decreases population turnover necessary for rapid evolution. Consequently, maladaptation increases and population size decreases faster than the species' ranges. Yet, evolutionary processes are at work and allow evolution to rescue maladapted populations if the pace of climate change were to slow down or stop. Our framework, by explicitly incorporating species demography and evolutionary potential, highlights how local eco-evolutionary processes translate into changes in species range.

## Results

**Niche projections and eco-evolutionary forecasting.** Since SENMs assume no explicit demography, projected species' responses to climate change are immediate and show an overall contraction of species' ranges (Fig. 1 and Supplementary Fig. 4). In contrast, DEEMs predict an early expansion due to colonization outside the initial range, followed by a decrease until 2150, which produced a signal of extinction debt[22] (Fig. 1 and Supplementary Figs 5 and 6). The decline of range sizes in DEEMs is caused by increased maladaptation to locally changing climatic conditions. First, the increase in the suitability of unoccupied patches, in addition to the increase in

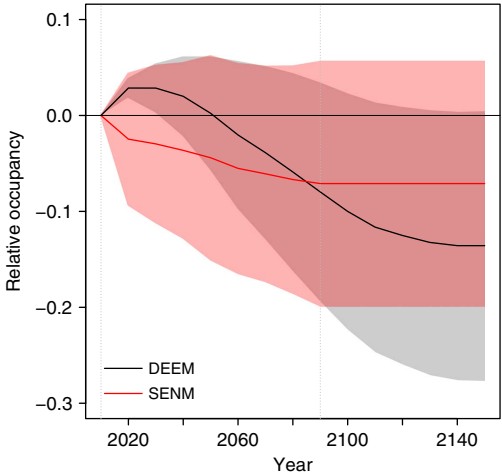

**Figure 1 | Predicted change in occupancy as a function of time.** The change in occupancy is measured relative to the initial occupancy in 2010, and is calculated as (number of cells occupied in year $t$ − number of cells occupied in 2010)/number of cells occupied in 2010. Lines and coloured areas: mean and s.d. over all species, grids, climate scenarios, selection strengths and mutation rates investigated.

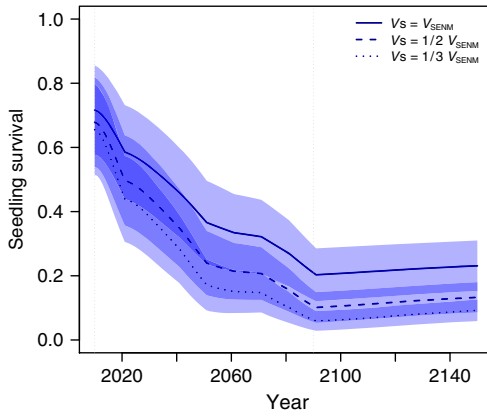

**Figure 3 | Seedling's adaptive survival rate as a function of time for three strengths of selection ($V_s$) in the DEEMs.** The seedling-adaptive survival rate is calculated from equation (2) and does not take into account competition with adults. $V_s$ of a trait is measured as a proportion of the variance $V_{SENM}$ of its corresponding environmental variable in the initial niche as predicted by the SENMs (see Supplementary Table 1). Lines and coloured areas: mean and s.d. over all species, grids, climate scenarios and mutation rates investigated. Solid line: $V_s = V_{SENM}$; dashed line: $V_s = 0.5 \times V_{SENM}$; dotted line: $V_s = 0.33 \times V_{SENM}$. The strength of stabilizing selection decreases with $V_s$ (equation (2)).

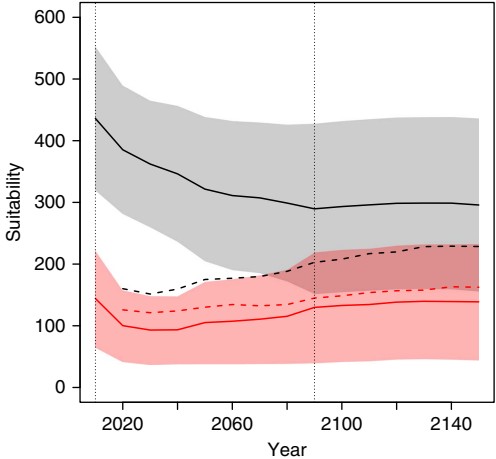

**Figure 2 | Mean site suitability in the DEEM simulations.** Suitability is predicted by the SENMs and corresponds to the probability of presence × 1,000. Solid lines show the suitability in occupied (black) and unoccupied sites (red) in the DEEMs. The dashed lines show the suitability in the sites colonized (black) and extinct (red) relative to the population state in 2010. For the DEEM predictions after 2090, we used the suitability from the SENMs in 2090 since the climate was assumed constant between 2090 and 2150.

the suitability of newly colonized patches, demonstrates that migration allows the colonization of suitable patches but is not sufficient to track the displacement of the initial niche (Fig. 2). Second, at the onset of climate change, local site suitability decreased (Fig. 2), causing a rapid decline of seedling survival (Fig. 3). However, adaptive processes do occur, as shown by the rebound in seedling survival (Fig. 3) and population size (Fig. 4) after 2090 (see also Supplementary Fig. 7).

**Factors influencing the changes in species range.** The decrease in range sizes occurs sooner in species with reduced population size (lower carrying capacity; Fig. 4a), such as *Campanula* and *Dianthus*, than in species with large population size

(*Festuca*; Fig. 4a), consistent with classical theoretical predictions[24]. Variations in population sizes are further correlated with climate specialization (Supplementary Table 1) and range fragmentation (for example, *Campanula*; Supplementary Fig. 3). Range loss is more pronounced in species facing larger temporal shifts of local conditions relative to their phenotypic trait variation within populations (Fig. 5), which causes a faster decrease in population sizes (for example, *Campanula*; Supplementary Fig. 8). Reduced population size decreases the ability of selection to drive evolutionary adaption to the environment and increases random genetic drift (for example, ref. 33). Further, populations below ~100 adults incur higher extinction risks due to demographic stochasticity (for example, ref. 34; Supplementary Figs 5, 6 and 8). The joint decrease of trait genetic variation (Supplementary Figs 9–11) and population size creates an extinction vortex (Supplementary Fig. 12) where local populations cannot escape a regime of genetic and demographic stochasticity despite the stabilization of climatic conditions after 2090. This generates an extinction debt. Yet, populations that escape that regime can quickly rebound and adapt to the local conditions (Fig. 3). This differs from the extinction debt as predicted by hybrid models combining niche projections and demography without evolution (for example, ref. 22), where the contraction of the range lasts after climate stabilization due to population persistence in unsuitable environments without possibility of recovery.

**Strength of selection and adult survival rate.** The overall effect of climate change on species' range is modulated by the strength of selection on seedling survival and by the adult survival rate (Fig. 6 and Supplementary Fig. 13). Stronger selection on seedlings leads to a faster decline of their survival with climate change (Fig. 3), resulting in a fast decrease in population size (Supplementary Fig. 8). More interestingly, adult survival has antagonistic effects on the dynamics of occupancy and maladaptation. Higher adult survival leads to longer persistence of adults and thus slows the decrease in species' ranges (Fig. 6). On the other hand, lower adult survival strongly jeopardizes

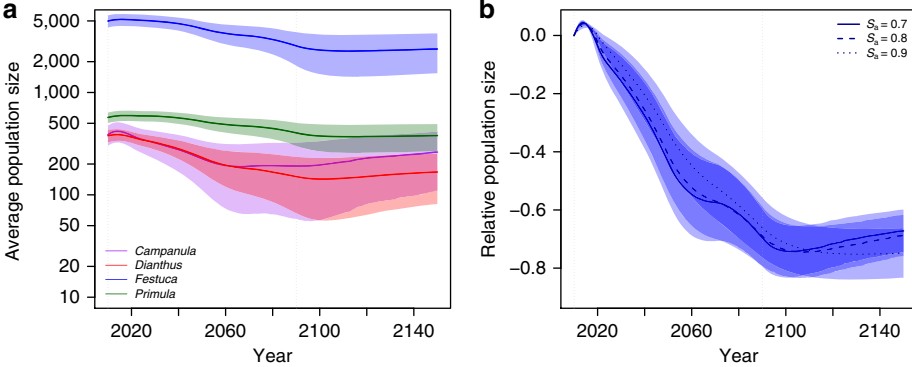

**Figure 4 | Within-site adult population size as a function of time in the DEEMs.** (**a**) Number of adults; (**b**) relative to adult number in 2010. Solid line and coloured area: mean and s.d. across all simulations.

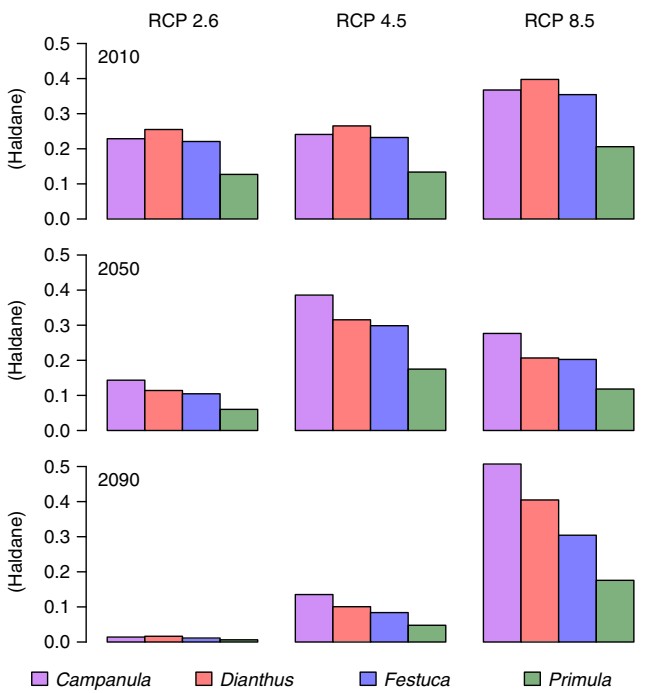

**Figure 5 | Rate of environmental change in occupied sites in the DEEMs.** The environmental change is measured in units of phenotypic s.d. (Haldane) for the four species and three different years within the period of climate change (2010–2090) for Bio 1. Each bar is the average for a given RCP scenario over all grids and parameter values in each species.

species persistence (Fig. 6) but favours population turnover, which in turn improves the chances of evolutionary rescue by adaption (Fig. 4b).

**Changes in age structure.** The age structure of the population determined the dynamics of the response to climate change. At the stable age structure, the population was primarily composed of reproductive adults, followed by seedlings and pre-reproductive adults, respectively (Fig. 7). Maladaptation to new local conditions decreased the frequency of pre-reproductive adults (Fig. 7) because of a decrease in seedling survival and thus of seedling recruitment. Furthermore, long-lived adults persisting in populations during climate change competitively restricted seedling recruitment and produced seedlings increasingly maladapted to the new climate. Therefore, new individuals did not replace senescing adults, which decreased the adult frequency relative to the seedlings as population size dwindled. After the climate stabilized, the frequency of pre-reproductive adults increased again (Fig. 7), showing that better adapted seedlings were recruited and hence average maladaptation decreased (Fig. 3). An immediate increase in population size accompanies the recruitment of these adapted individuals (Fig. 4 and Supplementary Fig. 8). The lifespan of adults is thus a key parameter for adaptation to the relatively fast climate change, and rapid shifts in age structure, a hallmark of growing local maladaptation.

**The role of local adaptation.** One key assumption of DEEMs is that populations are locally adapted[35]. By contrast, SENMs assume that a single ideal phenotype exists, which represents an average of the eco-climatic values in which the species is found. Occurrence probabilities provided by SENMs can, in principle, be interpreted as performance indicators of that average, globally adapted, phenotype (for example, ref. 22 but see ref. 36). To evaluate the importance of local versus global adaptation, we ran simulations in which individuals had a single genotypic value corresponding to the average environment in occupied patches. This genotype was not allowed to evolve but was allowed to disperse. We found that species occupancy rates decline faster in the absence of local adaptation and evolution when SENMs predicted a degradation of local conditions (*Campanula, Dianthus,* and *Festuca*; see Supplementary Figs 4 and 14), but increased faster in the reverse situation (*Primula*; Supplementary Figs 4 and 14). Local adaptation and the presence of a larger set of possible genotypes across a species' range thus buffer against rapid climatic changes outside of the original niche by providing more genetic variation to track the shifting climate, although imperfectly. On the other hand, when an amelioration of local conditions towards the centre of the niche occurs (as for *Primula*; Supplementary Fig. 15), phenotypic diversity maintained by local adaptation is a handicap because more populations become maladapted. Yet, local population sizes were always smaller in the single genotype simulations (Supplementary Fig. 16), suggesting the occurrence of a cost of being a generalist.

**Discussion**
Our results suggest that, under changing climatic conditions, a long lifespan of adults has equivocal effects on the fate of species. On the one hand, long adult lifespan limits the adaptive capacity of local populations (see also ref. 32), on the other hand it allows long-term persistence in unsuitable sites (for example, *Festuca*). In our alpine case study, long lifespans, together with limited dispersal capacities, substantially delay the range loss of less

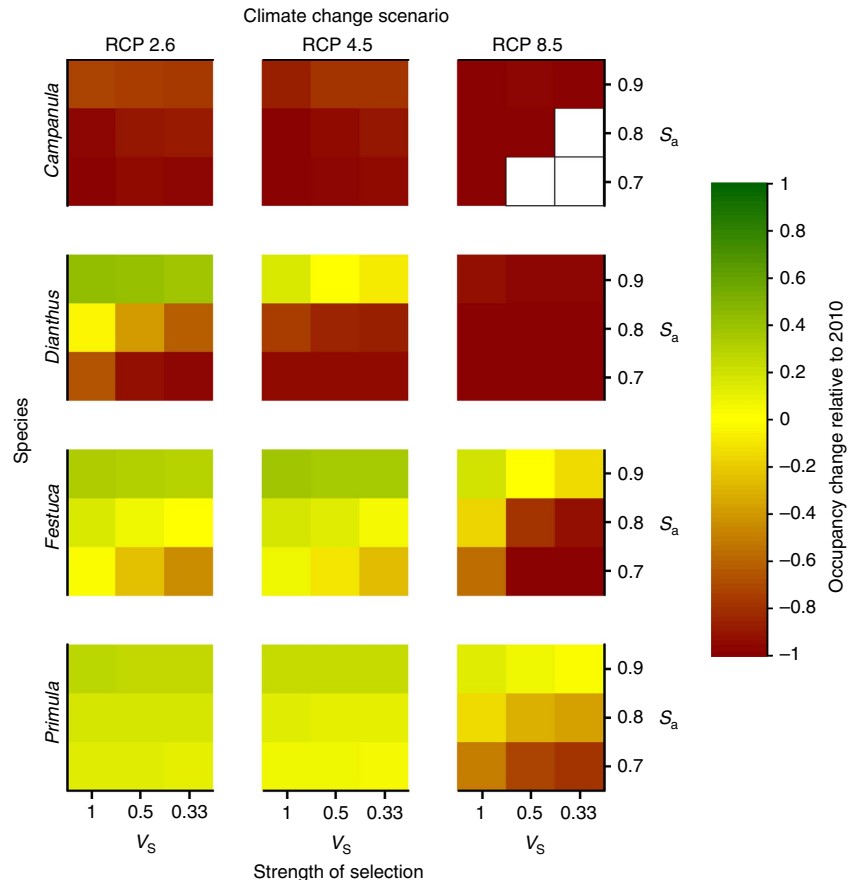

**Figure 6 | Variation of species' ranges recorded in 2150 relative to their initial state at the onset of climate change in 2010 in the DEEM.** The values shown are relative grid occupancy (number of occupied cells in 2150 relative to 2010) for each combination of relative strength of selection ($V_S$), adult survival ($S_a$), climate change scenario (RCP), and species, averaged over all grids in each species and for the three mutation rates. Warmer colours represent decrease of occupancy while cooler colours represent increase of occupancy. The averages are taken only for those simulations where at least one cell remained occupied in 2150. Empty cells correspond to cases with complete extinction of all populations in all grids.

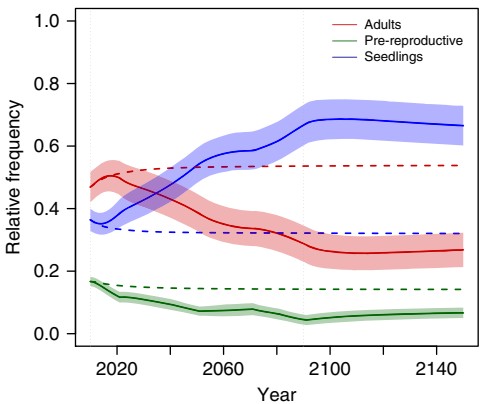

**Figure 7 | Age structure of the population within sites as a function of time in the DEEMs.** Solid lines and coloured areas: mean and s.d. across all simulations with climate change. Dashed lines: mean across all simulations with a constant climate. Red: adults; green: pre-reproductive adults; blue: seedlings.

specialized and more abundant species (sometimes beyond the twenty first century in our simulations, for example, for *Festuca* and *Primula*) corroborating the 'extinction debt' proposed by Dullinger *et al.*[22]. However, the slow decline in occupancy hides a rapid loss of local adaptation and a rapid decrease in

population density as the climate changes. Adaptive recovery nevertheless occurs once climatic conditions stabilize, revealing that adaptive tracking is active during climate change. The evolutionary recovery is favoured by shorter adult lifespan (as when adult survival is lower) because it causes faster recruitment of seedlings surviving the test of natural selection. This result suggests that slowing down the pace of climate change will increase the odds of evolutionary rescue, at least in relatively abundant species.

A major implication of our results is that changes of species ranges may be poorly connected to the state of local populations because local processes, both evolutionary and demographic, occur faster than changes to species ranges. Hence, we suggest that the monitoring of local population sizes is a more sensitive indicator of the future development of a species' range, that is, whether it is likely to shrink or to expand, than larger-scale presence–absence surveys. Our results suggest that the size of local populations can quickly fall below a threshold value where stochasticity may over-rule even favourable deterministic trends. Consequently, even relatively widespread Alpine species can quickly disappear because of small local population sizes and poor connectivity between populations.

The DEEM framework that we propose here is a powerful tool to predict future species range in response to climate change and represents an attractive future research avenue. This framework allows the tracking of demographic and evolutionary dynamics in populations and accounts for the stochasticity of these processes.

We showed that the understanding of how eco-evolutionary processes affect local populations is determinant to predict changes of a species' range. The importance of demography in perennial plant species has been demonstrated previously using a so-called hybrid model that couples SENM projections with a parametric demographic model and spatially explicit dispersal[22]. That model predicted slower decrease in species' ranges across the entire Alps compared to SENM projections and revealed the existence of an extinction debt caused by the persistence of individuals in deteriorated environments and limited dispersal capacities. Our results are consistent with these previous findings, but further highlight the feedback between demography and evolutionary dynamics, which often caused a larger negative response to climate change than the SENM projections or the hybrid model would predict, especially under worst-case scenarios (for example, representative concentration pathway (RCP) 8.5, strong selection and low adult survival, see Supplementary Fig. 17). Furthermore, a recent hybrid model that coupled SENMs with a parametric evolutionary model obtained slower losses of species' range of Drosophilid species in Australia than predicted by SENMs under climate change[16]. That model, however, did not account for demography or any stochastic process (for example, drift and mutation), and imposed extinction on populations that failed to provide an evolutionary response above a fixed threshold. An extinction debt would thus not develop either because populations have enough trait variation to provide an evolutionary response or go extinct, as when the trait under selection reaches its physiological limit. By allowing the mean of the initial niche, summarized by the maximum thermal tolerance in ref. 16, to continually evolve, parametric evolutionary hybrid models might reach the simple conclusion that populations will do better than expected under SENM projections unless heritable variation is too limited or evolutionary limits are reached. This last approach has been applied to Drosophila flies, with short life cycle relative to the timing of climate change and probably large population size (see also ref. 31), where considering demographic details may be less relevant to predict the future species' range. How much modelling choices regarding demographic and genetic details matter for predicting species range may depend on the characteristics of the focal species. Nevertheless, our results strongly suggest that details of the evolutionary and demographic processes can deeply affect the interpretation and understanding of current and future species range variations.

Our results, as a modelling exercise to project the future range of actual species in their environment, rely on the data available and model assumptions. First, we selected ecologically relevant and relatively independent bioclimatic variables and consequently assumed that the quantitative traits underlying adaptation to these variables are independent (Methods). Yet, most bioclimatic variables are strongly correlated (Supplementary Fig. 18), possibly imposing correlated selection on phenotypic traits and influencing the speed of adaptation to the changing environment (for example, ref. 37). Similarly, genetic correlations among those phenotypic traits may further act as constraints on adaptation if climate-related selection acts against genetic trade-offs or imposes correlated selection in other phenotypic traits that do not vary with the climate[37,38]. Indeed, other factors than bioclimatic variables might be relevant to model Alpine species ranges at a detailed local scale[39]. Our simulation framework would allow incorporating this level of genetic details, accounting for genetic trade-offs among traits, but it is left to explore in future work.

Second, we modelled the life cycles to the best of the data available for the focal species, with constraints on the computational resources available. Most importantly, we assumed that selection exerted by climatic variations mostly affects seedling survival, consistent with previous studies[40,41]. In preliminary simulations, we tested the effect of selection on adult fecundity (Supplementary Figs 19 and 20), and found that climatic variations do not affect species range under this assumption. This result is consistent with a preliminary analysis of the sensitivity of the population growth rate to variations of adult fecundity in a deterministic demographic model (described in Supplementary Methods), and shows that climate change does not decrease fecundity enough to lead to negative growth rates. The range of Alpine plant species has been observed to decrease, most likely due to climate warming (for example, ref. 42), suggesting either that our assumption that the environment affects mostly seedling survival (on which the population growth rate is more sensitive than fecundity) is reasonable, or that selection is much stronger than that applied in our model. Alternatively, using a Gaussian selection function with s.d. from niche models might be inaccurate, and truncation selection, for example, due to lack of cold temperature necessary to germinate, can be more appropriate to investigate this case (but not implemented in our model). Further, community composition has been proposed to be a better predictor of adult fecundity than environmental variables[43]. In any case, the identification of the life stages primarily affected by climate variation is required to provide accurate predictions of how species would be affected by climate change. To this end, high-quality database on plant demography as compiled in the COMPADRE database[44] with precise information on species distribution would be particularly suitable to the present framework. Our main results are nevertheless qualitatively robust to the sensitivity analysis we performed on the key parameters for which no precise estimates were available, such as strength of local stabilizing selection (Fig. 3 and Supplementary Fig. 5), adult survival rate (Fig. 6) and mutation rate of the quantitative loci (Supplementary Fig. 7).

In summary, our new approach highlights how eco-evolutionary dynamics at the population level may translate into variation of species' ranges under a warming climate. We show that the persistence of established long-lived adults combined with low dispersal capacities delays the contraction of Alpine plant species ranges while local maladaptation immediately increases. In addition, we demonstrate that short-term adaptation to climate change is unlikely in these species. These results imply that in long-lived species an extended phase of range stability may precede a rapid population decline when climate warms. During the stable phase, alterations of the population structure may be the only detectable effect of the changing climate. Demographic monitoring of local populations should hence become a standard component of biodiversity monitoring under climate change.

## Methods

**Species selection.** We performed a Hill–Smith analysis[45] based on 24 endemic alpine plant species from the species set in Dullinger *et al.*[22] in order to select four endemic species as different from each other as possible. The analysis was performed with the function dudi.hillsmith in R package ade4 version 1.7-2 (ref. 46). The principal component axes did not discriminate strongly species variation: the first principal component axis explained ∼22% of the variance and the second explained ∼17% of the variance. Supplementary Fig. 1 shows the mapping of the selected species and factors along the first two principal component axes. On the first axis, we selected *C. pulla*, which produce many dispersive seeds and *D. alpinus*, which has a high clonal growth but is a poor disperser. On the second axis, we selected *F. pseudodura*, which is a dominant species but has a low germination rate and *P. clusiana*, which has a high germination rate but is subdominant.

**Studied areas.** The species that we selected are endemic in the Austrian Alps (Supplementary Fig. 2). We restricted our analysis to 15 grids of 32*32 cells (250 m × 250 m squares, or 64 km²) located in the northeastern Alps (Supplementary Fig. 3).

**Static ecological niche models.** We fitted niche models for the four species by relating true presence and absence data from 2,386 vegetation plots to one soil and two bioclimatic variables downscaled to a 250 m spatial resolution. We used the averaged climate between 1970 and 2005 for the current predicted range and the climate predicted by the RCP scenarios for range projections to 2090 (see Supplementary Methods). The bioclimatic variables were chosen to (i) be as independent as possible in a principal component analysis using the 19 bioclimatic variables proposed by WorldClim, (ii) contain information about temperature and water availability and (iii) be ecologically relevant. With these criteria, we chose the mean annual temperature (Bio 1), the annual amount of precipitation (Bio 12) and the percentage of carbonate in the bedrock.

The SENMs were performed with Biomod2 R package[47]. An ensemble of forecasts of niche model models was obtained for each one of our selected species using the presence–absence and the environmental data (climate and soil) explained above. The ensemble included projections with generalized linear models (with linear and quadratic terms selected through a stepwise procedure), generalized additive models (with a maximum degree of smoothing of 3), surface range envelopes and random forests. Models were calibrated for the baseline period using 70% random sample of the initial data and evaluated against the remaining 30% data, using the True Skill Statistic (TSS). This analysis was repeated five times, thus providing a fivefold internal cross-validation of the models. Each calibrated model was used to forecast the probability of occurrence of the species on the Austrian Alps under both current conditions and climate change scenarios. Instead of using the five forecasts for each of the four algorithms, we instead combined them into a single ensemble forecast with an average weighted by the relative performance of the models extracted from the TSS analyses (models with a TSS lower than 0.3 were discarded). Finally, a species was considered present in a patch if its occurrence probability was above a threshold value that maximizes the final TSS statistics.

**Dynamic eco-evolutionary models.** Our DEEM framework is built on the individual-based, stochastic and forward-in-time evolutionary simulator Nemo[48], which we modified for complex life cycles.

The phenotype of an individual is composed of three polygenic quantitative traits each corresponding to one of the three environmental variables. Each trait is under the control of 10 unlinked and additive diploid loci bearing pleiotropic mutations. Mutations per allele occur at rate $\mu$. Each new mutation is drawn from a three-dimensional Gaussian distribution with the mean of zero and equal variance $v$ (set to $v = 0.05$ in all simulations) for each trait with no correlation. The three random mutational effects are added to the existing allelic values at a locus to yield the new allelic value of each trait (that is, continuum-of-allele model[49]). The genotypic value of a trait is the sum of all alleles contributing to that trait. The per-trait mutational variance generated in each patch equals $2L\mu v$, with $L$ the number of loci, and is not dependent on the initial trait value. Finally, the trait's phenotype of an individual is the sum of its genotypic value and a random environmental component drawn independently for each trait $k$ in a centred Gaussian distribution with variance $V_{E,k}$ set such that the average trait heritability is $h^2 \sim 0.3$ (see Supplementary Table 2).

We assumed hermaphroditic individuals with four life stages: seeds, seedlings, pre-reproductive adults and adults. The life cycle in the simulations starts with mating and seed production, followed by seed dispersal, aging of adult and pre-adult individuals, seed germination and survival in the seedbank, clonal reproduction (produces new seedlings from adults and pre-adults), seedling competition (density-dependent regulation) and seedling viability selection. The life cycle was optimized to minimize the simulation time. Census is done after seedling competition and before seedling selection. The model tracks the number of individuals in each stage for each year. The yearly demographic recursions and probabilities of transition between stages are provided in Supplementary Methods and Supplementary Table 3. The seeds of Campanula and Festuca can only germinate after 1 year (that is, no seedbank), whereas they can germinate after up to 5 years from the seedbank in case of Dianthus and up to 7 years in case of Primula[50]. Seeds that have not germinated within this maximal time of seed persistence are discarded. After germination, seedlings either survive selection or die. Following the seedling stage, individuals mature for 2 years before entering the reproductive adult stage. Reproductive adults can survive for several years, with a probability $s_a$. We performed a sensitivity analysis on this parameter because no precise estimates were available for the focal species. We considered three values (0.7, 0.8 and 0.9; that is, adult survival expectancies within (3–10) years) according to estimated adult survival rates in species with similar ecology[51–53].

Seedlings (produced by both sexual and clonal reproduction) suffer competition from adults following a Beverton–Holt function. The probability that a seedling survives competition in patch $i$ at time $t$ is given by

$$c_{i,t} = \frac{1}{1 + k_c N_{i,t}^{Adults}},\qquad(1)$$

where $k_c$ is the competitive weight of an adult individual (Supplementary Table 4) and $N_{i,t}^{Adults}$ is the number of adult individuals in patch $i$ at time $t$. Equation (1) is motivated by the fact that seedlings mostly compete against adults previously established in favourable microhabitats. The transition rates between stages and $k_c$ determine the carrying capacity at demographic equilibrium. We set

$k_c$ to keep the carrying capacity of each species at the order of magnitude of the estimated adult population size from Hülber et al.[50] and to conserve the relative difference in abundance between species. We set the maximum number of adults to 1,000 for Campanula, 300 for Dianthus, 500 for Primula and 5,000 for Festuca in a single perfectly adapted population. The carrying capacity of Festuca was kept at the computationally manageable maximum. Finally, note that the observed population size is below these values in the simulations because of the migration and mutation loads (that is, reduction in the mean population fitness caused by maladapted individuals).

Viability selection occurs on seedling survival. In alpine plant species, seedlings are particularly sensitive to temperature and moisture (for example, refs 40,41). We model Gaussian selection where the survival probability of an individual with phenotypes $\mathbf{z}$ in patch $i$ at time $t$ is:

$$W_{i,t}(\mathbf{z}) = \exp\left[-\frac{1}{2}(\mathbf{z} - \boldsymbol{\theta}_{i,t})'\mathbf{W}^{-1}(\mathbf{z} - \boldsymbol{\theta}_{i,t})\right],\qquad(2)$$

where the prime indicates transpose. $\boldsymbol{\theta}_{i,t}$ is the vector of optimal phenotypes in patch $i$ at time $t$ (that is, given by the environment at time $t$) and $\mathbf{W}$ is the pattern of multivariate selection. We assume that selection is uncorrelated between traits, so that $\mathbf{W}$ is a diagonal matrix with variances for each trait $k$ (diagonal elements) given by $V_{s,k}$. The strength of natural selection is thus inversely proportional to $V_{s,k}$: the lower $V_{s,k}$ the stronger the selection.

To initialize the simulations, the trait values of all individuals in a site (a grid cell) are set equal to the environmental values within that patch before climate change. Starting genetic variation is added by generating random mutations to each allele. Hence, populations are initially locally adapted but with large genetic and environmental variation around the local site optimum. Migration further adds genetic variation within sites (see the burn-in simulations below). To set the selection strength on each trait, we used the variance of each environmental variable in the sites predicted to be occupied by the SENMs, $V_{SENM}$, as a reference (that is, the niche width). We tested several strengths of selection, assuming that $V_{SENM}$ is the lowest selection strength that can be expected (Supplementary Table 1). In addition, we compared the results of the DEEMs with local adaptation to simulations initiated with a unique genotypic value calculated as the average environmental value of the sites predicted to be occupied by the SENMs within each grid. The unique genotype was calculated for each grid to avoid a grid effect.

Simulations were performed in two phases. First, we simulated a burn-in period of 5,000 generations to allow populations to reach their evolutionary mutation–selection–migration and demographic equilibria. Initial individual distributions in each grid were based on suitability projections given by the niche models using a prevalence threshold. During burn-in simulations, we prevented the species to expand geographically by defining an additional ceiling-regulation event in the life cycle based on a carrying capacity set to zero in sites predicted unoccupied by the niche models and to a large value (8,000, that is, above the population size expected at equilibrium) in the other sites. A large carrying capacity allowed populations to reach their demographic equilibrium while preventing them to explode (thus, reaching computational limits) in the initial exponential growth phase. Species ranges can indeed grow indefinitely in the case of a static environmental gradient when the genetic variance is not constrained[26,54]. Burn-in populations were saved and used in the second phase to simulate shifts in local climatic conditions using different climate change scenarios.

We ran the simulations for years 2000–2150. We used climatic projections from three RCP scenarios between 2010 and 2090 (Supplementary Methods). The climate remained constant after 2090. We performed all simulations for three allelic mutation rates ($\mu = 0.01$, 0.001 and 0.0001), strengths of viability selection ($V_s = V_{SENM}$, $V_s = 0.5 \times V_{SENM}$ and $V_s = 0.33 \times V_{SENM}$) and adult survival rates (0.7, 0.8 and 0.9), with 10 replicates for each combination of species, spatial grid, RCP scenario, $\mu$ and $V_s$. For each simulation, the following data were saved before selection in intervals of 10 years from 2000 to 2150: individual number per site and age class, average and variance of genetic and phenotypic values for each trait in occupied sites, and average seedling fitness values in occupied sites.

**Data availability.** All models were run using the BIOMOD package[47] in R (the complete script is available here: https://github.com/DamienGeorges/endemicitysdmhub/blob/master/SCRIPTS/2_species_niche_modelling.R). The extended code of Nemo[48] and the corresponding input simulation files will be available upon request before publication of the version of Nemo we used in this study. A GitHub repository will then be updated: https://github.com/oliviercotto/NCOMMS-16-18804A.

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

## Acknowledgements

O.C., M.S. and F.G. acknowledge support from the Swiss National Science Foundation (grant PPOOP3_144846). W.T. acknowledges support from the European Research Council under the European Community's Seven Framework Programme FP7/2007–2013 Grant Agreement no. 281422 (TEEMBIO). G.K. and S.D. were supported by the Austrian Science Foundation (project 'Who Is Next' I-1443-B25). J.W. and S.D. acknowledge funding from the University of Vienna's rectorate. Parts of the simulations were performed on the CIMENT HPC infrastructure (https://ciment.ujf-grenoble.fr), supported by the Rhône-Alpes region (GRANT CPER07_13 CIRA: http://www.ci-ra.org) and France-Grille (http://www.france-grilles.fr). The University of Zurich, S3IT team (http://www.s3it.uzh.ch), provided further computational support and HPC resources. We thank M. Servedio, C. Fitzpatrick, J. Yeh, A. Ozgul, T. Bonnet and K. van Benthem for helpful comments on a former version of the manuscript.

## Author contributions

O.C., F.G. and W.T. designed the study. D.G. and W.T. calibrated and fitted all niche models. S.D., J.W. and G.K. provided the data for plants and processed the climatic data. O.C., F.G. and M.S. provided C + + code for the individual-based simulations. O.C. and F.G. performed the individual-based simulations. O.C. analysed the data for plant traits and the niche model results. O.C. and F.G. analysed the results from the individual-based simulations. O.C. and F.G. wrote the paper, with substantial help from all co-authors.

**Additional information**

**Competing interests:** The authors declare no competing financial interests.

