## [Peer Review File · Nature Communications]

Reviewers' Comments:

Reviewer #1 (Remarks to the Author):

The work by Cotto and collaborators “Slow eco-evolutionary response of alpine plant to climate warming” builds climatic projections for the spatially explicit viability of four plant species from the Alps of, as the authors state, high longevity, and various modes of life history before various climatic scenarios. The authors find that the persistence of adults in non-suitable habitats allows the species’ populations to persist beyond what models not incorporating demographic information predict. This supposedly would result in a buffer against local and eventually total extinction by all four species, though at various rates, which all would eventually result in demographic failure.

I have found the main paper well written overall, but that critical parts of the modeling exercise should be more detailed in the main paper (and also in the extended methods!). I see no problem with the SDMs, but I see several alarming assumptions that were not tested, and even if they had been tested, the results should be taken with a lot more care than they are explained in the paper. Most of the criticisms below are on the weakness of the comprehension and treatment of the demography of the species that the authors show in the main paper and extended methods:

- The authors say these are long-lived species, but information about mean life expectancy (which can be calculated with one line of code, see below) or age of cohort at 99% mortality, is never provided. I have calculated these rates (ignoring clonality, see below) and they result in mean life expectancies of 2-4 years for life cycles conditional on entry from the seed stage... or about 20 for adults. Clearly these are not long-lived species, or at least not to the extent that the authors have sold this manuscript.

- The assumption of survival 0.95% for adults in the extended methods renders this manuscript very difficult to evaluate. Any life history expert would tell you that the vital rate for long-lived species where one should not make assumption, but actually get good estimates (with variances around them, which here are not presented), are precisely the ones that the authors have pulled out of thin air: adult survival. I am concerned that the information on the demography presented for each species makes no sense... they are all too close to each other, and in some cases some raise population growth rates (as calculated by me in R) of 1.72 for *Campanula* or 1.44 for *Primula*, which are really high for long-lived species... but see below that these estimates must be even greater when including clonal rates

- The authors state that each species has a given rate of clonal reproduction, but they don't tell where in the matrix population model this should go. Without this info, I can't verify the longevity of the genets (if that's what the mean, and not the ramet), and of course the population growth rate estimates from the point above would be even greater... but that exact value would depend on where the numbers go in the matrix model. The assumption of taking the median for this (L. 504) negates decades of demographic advancements on stochastic population models.
- The info presented in the matrices implies no seedbank exists for any of these species. Can the authors clearly show evidence that this is the case? The existence of a seedbank can drastically affect the population dynamics of plant species...
- If I were to do this analyses, I would do it with species for which a lot of demographic information (in situ) has been collected and published for many years and locations (and the ideal repository for that already exists – www.compadre-db.org), rather than pull from thin air demographic information. I would also perhaps include an expert in demography to help with the stochastic modeling of vital rates and potentially with the dispersal kernels – the latter might have been done correctly, but with the limited info at hand the authors have provided, I cannot state further.

There are many other points I have made in the PDF. Most of them have to do with my inability to objectively evaluate aspects of the model because not enough details have been provided (e.g. dispersal information, soil information, detail on True Skill Statistic), or because figure legends are not fully coherent.

PS: Code (from Caswell 2001) to estimate mean life expectancy, where U is the matrix without reproductive rates:

```
colSums(solve(diag(4)-U))[1]
```

Reviewer #2 (Remarks to the Author):

Review of NCOMMS-16-18804 “Slow eco-evolutionary response of alpine plants to climate warming”

Jorge Soberón, University of Kansas

This paper describes the application of a modeling exercise that combines niche modeling with dispersal and natural selection on the characters defining the niche. It is based on well-studied alpine plants, and presents interesting results, both in terms of fundamental ecology, and of

relevance to adaptation to climate change. The results are novel (but see below) and general, and certainly new for the plants and the ecosystem in question. Although many antecedents to this paper can be found in the theoretical literature, the results in the context of niche modeling and climate change are most likely unappreciated. Moreover, the described methodology should become a welcome addition to the toolbox of students of ranges of distribution under climate change. I believe the paper should be published.

Having said that, I still have a few comments. The main one is related to the lack of mention to the theoretical antecedents of the work. There are quite a few papers highly relevant to this work that are not mentioned, and they should, since the authors are essentially exploring range dynamics under climate change, a topic with a rich literature. Their discussion (mainly lines 96-119) will gain clarity and force if some previous results were actually incorporated: Holt and coworkers have discussed these problems since long ago (Holt & Gomulkiewicz, 1997a; Holt & Gomulkiewicz, 1997b) showing that, under very general conditions, subpopulations outside the limits of the niche will go extinct before natural selection can rescue them. This result does not depend on any spatial structure, but it can probably be useful to understand the results in the manuscript.

Other factors hindering the expansion of areas of distribution (Kawecki, 2008; Sexton et al., 2009) include gene flow from populations adapted to conditions in the core of niche space to populations outside or marginal to the niche. This was first mentioned by Haldane (Haldane, 1956) half a century ago and is very relevant to the work under review. Recent theoretical results (García-Ramos & Kirkpatrick, 1997; Kirkpatrick & Barton, 1997) confirm those early hypothesis. In the simulation in the MS this effect is almost certainly taking place but it is not mentioned. I suggest that using the results will add value to the discussion.

Finally, in a rather obscure paper (and in Spanish...

<http://www.misclaneamatematica.org/Misc49/4906.pdf>), Soberon & Miller (Soberón & Miller, 2009) report results of a simulation of natural selection on a two dimensional niche, on a grid of populations, with and without spatial covariance on the environmental factors. They show that the spatial covariance of environmental values (and therefore of selective pressures) is determinant of the details of the evolutionary dynamics of the niche, and of total population size. Again, I suspect that the simulation described in the MS being reviewed is also affected by spatial covariance but the authors do not include this point in their discussion. It may be interesting simply to present variograms of the environmental values as climate changes, and discuss whether climate is moving towards higher or lower spatial covariance (higher covariance facilitates evolution, lower covariance slows it down).

Many of the above results can be used to compare, contrast and explain the results of the paper's simulation.

There are several little details:

- 1) It is rather puzzling to see the paper justified (L41-43) by reference to documents of bureaucratic bodies. Maybe cite chapter and verse of the reports of IPBES that are relevant to a paper about the evolution of range-sizes? I rather would like to see reference to some of the conceptual papers by Holt, Kirkpatrick, Kawecki or any of those, but this really is up to the authors.
- 2) L55-57. The factors mentioned here are precisely the factors that the papers quoted above, and not referred to, analyze!
- 3) Line 68. Substitute “real” by “realistic”?
- 4) L436-437. Downscaling WorldClim data to 100 meters is probably extremely unadvisable, since at 100 meters weather is really becoming microclimate, influenced by topography (even microtopography), habitat, wind, soil... none of which is included in the original WorldClim database. Downscaling without recalculating using such microclimate-relevant parameters is essentially a numerical exercise, with no real value. Please justify this step.
- 5) L449-450. Seems to me that the interval are of 14 years, not 17... Please check or clarify.
- 6) L480-482. What algorithm/s was/were used by Biomod2? With what settings? Presence-absence or presence only? An ensemble? Of what? There are no details whatsoever in this very important part of the modeling process. The fact that the authors use (L483) “probability of occurrence of a species” suggest they were using some variety of logistic regression with presence-true absence data. If this is not the case, then correct this as well.
- 7) There is no description of the settings of the simulator Nemo. This is a highly sophisticated program, with many parameters. I suggest that the details of the simulation, including the external text files with parameters are added in the supplementary materials.

García-Ramos, G. & Kirkpatrick, M. (1997) Models of adaptation and gene flow in peripheral populations. *Evolution*, 51, 21-28.

Haldane, J. B. S. (1956) The Relation between Density Regulation and Natural Selection. *Proceedings of the Royal Society of London. Series B, Biological Sciences*, 145, 306-308.

Holt, R. D. & Gomulkiewicz, R. (1997a) The evolution of species niches: a populations dynamic perspective. *Case Studies in Mathematical Modelling: Ecology, Physiology and Cell Biology* (ed. by H.G. Othmer & F.R. Adler & M.A. Lewis & J. Dillon), pp 25-50. Prentice-Hall, Englewood Cliffs, NJ.

Holt, R. D. & Gomulkiewicz, R. (1997b) How Does Immigration Influence Local Adaptation? A Reexamination of a Familiar Paradigm. *The American Naturalist*, 149, 563-572.

Kawecki, T. J. (2008) Adaptation to marginal habitats. *Annual Review of Ecology, Evolution and Systematics*, 39, 321-342.

Kirkpatrick, M. & Barton, N. H. (1997) Evolution of a species range. *The American Naturalist*, 150, 1-23.

Sexton, J. P., McIntyre, P. J., Angert, A. L. & Rice, K. J. (2009) Evolution and ecology of species range limits.

Soberón, J. & Miller, C. P. (2009) Evolución de los nichos ecológicos. *Miscelánea Matemática*, 49, 83-99.

Reviewer #3 (Remarks to the Author):

This paper presents a modelling exercise using a novel expansion of niche based species distribution models, DEEM. Curiously enough this acronym is not spelled out in the text, but these models incorporate short-term evolutionary response, thus enabling examination of eco-evolutionary dynamics. The models are used to predict responses of four long-lived alpine plants to projected climate change over the coming 135 years (until the year 2150). The main conclusions are that these long-lived plants will have difficulties to adapt to climate change, eventually becoming more 'maladapted', and that their populations therefore will decline. The response is however delayed due to the plant's adult life-span. This will contribute to an extinction debt.

The main value of the study lies in the approach, which can be useful to predict species response to climate change, and in the sense that the predicted results can inspire more detailed studies of the eco-evolutionary dynamics of species (although for such studies short-lived plants would be more suitable).

Focusing on the specific results, on the other hand, the conclusion that long-lived organisms has a slow evolutionary response (in an absolute time-scale) is of course well-known, and will hardly surprise anyone.

As far as I can judge the modeling is technically accurate. My only technical question is whether the lack of fine-scale resolution may limit the applicability of the models. The climatic variables were downscaled to just 250 m resolution, meaning that microclimatic variation, which are now increasingly recognized as very important, not the least in alpine environments, are neglected.

However, as always in modelling the critical issue is the assumptions. I had a few remarks on these, which may necessitate some clarification.

The basic model (a static niche model) was based on one soil and two climatic variables. Many alpine plants are also strongly affected by grazing and/or management in high-altitude meadows. Now, I am not familiar with the chosen plant species, but would appreciate a comment on other factors, not included in the models, that may be important for these plants.

A key assumption is that populations are locally adapted. Thus, the environment changes, whereas the plants (due to their long adult life span) genetically do not. Over time, the plants become maladapted. But what is the basis for assuming that the plants were adapted in the first place? There are several reasons why they may not have been perfectly locally adapted. Firstly, previous climate has also changed; climate change occurs all the time. And if the generation time is long, these plants may over a couple of generations have experienced different climate regimes (e.g. the 'little ice age' between the 15th and 19th centuries). What is the rationale for assuming that the plants today are locally adapted to current climate? Secondly, as clonal plants have an erratic and sporadic recruitment, there might be very weak selection on the recruitment and juvenile phase. Thus, the surviving adults may just have been lucky, rather than equipped with the 'best' locally adapted genotypes. So, what would happen if the assumption of an initial local adaptation is relaxed? Does anything then at all happen related to eco-evolutionary dynamics?

In the model, selection operates only on juveniles. This may seem reasonable, but the risk is that two other aspects of selection are neglected. Selection on the adult phase may operate if fecundity varies in response to climate change. Given the rare and sporadic recruitment episodes in many clonal plants, fecundity may actually be important. In addition, for clonal plants one should not downgrade the potential importance of within-genet selection (which may occur in clonal organisms).

Reviewers' comments:

Reviewer #1 (Remarks to the Author):

The work by Cotto and collaborators "Slow eco-evolutionary response of alpine plant to climate warming" builds climatic projections for the spatially explicit viability of four plant species from the Alps of, as the authors state, high longevity, and various modes of life history before various climatic scenarios. The authors find that the persistence of adults in non-suitable habitats allows the species' populations to persist beyond what models not incorporating demographic information predict. This supposedly would result in a buffer against local and eventually total extinction by all four species, though at various rates, which all would eventually result in demographic failure.

In addition to this comment, we would like to add that our simulations, along with explicit demography, incorporate evolutionary dynamics and the associated feedbacks. This has scarcely been done for predicting species responses to climate change. An important outcome of incorporating both demography and evolution is that the persistence of established adults acts as a demographic buffer but also slows down the evolutionary response of these species due to the persistence of maladapted adults, producing maladapted offspring and competing against seedlings for recruitment. The sensitivity of the results to adult survival that we performed for this new version confirms the importance of the above mechanism.

I have found the main paper well written overall, but that critical parts of the modeling exercise should be more detailed in the main paper (and also in the extended methods!). I see no problem with the SDMs, but I see several alarming assumptions that were not tested, and even if they had been tested, the results should be taken with a lot more care than they are explained in the paper. Most of the criticisms below are on the weakness of the comprehension and treatment of the demography of the species that the authors show in the main paper and extended methods:

- The authors say these are long-lived species, but information about mean life expectancy (which can be calculated with one line of code, see below) or age of cohort at 99% mortality, is never provided. I have calculated these rates (ignoring clonality, see below) and they result in mean life expectancies of 2-4 years for life cycles conditional on entry from the seed stage... or about 20 for adults. Clearly these are not long-lived species, or at least not to the extent that the authors have sold this manuscript.

We would like to thank reviewer for this comment. We indeed were not sufficiently precise here. All the species are long-lived in the sense that they are reproducing clonally and hence particular genets can occupy sites for a very long time (at least for centuries, e.g. Steinger et al. 1996). The numbers in the life table refer to adult ramets, however. A 20-year life time expectancy for an adult ramet is rather at the upper margin of values reported from species of similar habitats in the European Alps (compare e.g. Keller & Vittoz 2015 + refs in main answer). Moreover, and more important than whether a life expectancy of 20 years for adult ramets is long on an absolute scale, this lifespan of ramets is long relative to the velocity of climate change. Climate projections show that the mean annual temperature could increase enough in 20 years, so that organisms well adapted to the temperature at a given time can be substantially maladapted 20 years later.

- The assumption of survival 0.95% for adults in the extended methods renders this manuscript very difficult to evaluate. Any life history expert would tell you that the vital rate for long-lived species where one should not make assumption, but actually get good estimates (with variances around them, which here are not presented), are precisely the ones that the authors have pulled out of thin air: adult survival. I am concerned that the information on the demography presented for each species makes no sense... they are all too close to each other, and in some cases some raise population growth rates (as calculated by me in R) of 1.72 for *Campanula* or 1.44 for *Primula*, which are really high for long-lived species... but see below that these estimates must be even greater when including clonal rates

We agree with the referee that species-specific estimates on adult survival would be desirable. They are, however, available for very few species of the European Alps (and even for those they are likely specific to the particular populations sampled in these studies). To take into account the uncertainty about this parameter, we instead run a sensitivity analysis in the revised manuscript. The assessed range of values was based on demographic studies (Weppler et al. 2006, Kuss et al. 2008, Gonzalo-Turpin and Hazard 2009) of other high-mountain plants of the European Alps with similar ecology i.e. [0.7 – 0.9].

We moreover improved life-history schemes by including i) a seedbank, with species-specific seed survival times in the seedbank based on empirically evaluated relationships between seed persistence and seed morphology (Thompson et al. 1993, Schwienbacher et al. 2010); ii) different species carrying capacity (in Hülber et al. 2016, from Willner et al. 2012). Further, we tried to improve the precision of the other parameters by means of published data. Finally, we emphasize that the species also differ via their specific dispersal kernels.

We agree that the growth rates as extracted directly from the matrix are high. However, these rates are potential maxima and differ from those effective in the simulations. The parameter values in Hülber et al. (2016), on which part of our parameter estimates are based, are maxima from a number of individuals from different populations. During the burn-in simulations (prior to climate change), the populations reach their steady demographic states, where these potential

maximum rates are decreased by competition, migration and mutation loads, and where their actual growth is null.

- The authors state that each species has a given rate of clonal reproduction, but they don't tell where in the matrix population model this should go. Without this info, I can't verify the longevity of the genets (if that's what the mean, and not the ramet), and of course the population growth rate estimates from the point above would be even greater... but that exact value would depend on where the numbers go in the matrix model. The assumption of taking the median for this (L. 504) negates decades of demographic advancements on stochastic population models.

In the new version of the manuscript's Supp. Mat., we included a detailed explanation of the demographic recursions corresponding to the simulations (*Life cycle* section of the Supp. Mat.). Of course, the clonal growth rate increases the growth rate of the population but as explained above, the values in the tables do not take into account competition and maladaptation that affect clonal seedlings in our simulations. Effective growth rates in our simulations are hence much lower than those deducible from the input in the parameter tables. In addition, the demographic model in Nemo requires a single value for the clonal growth rate. The number of individuals produced is then drawn from a Poisson distribution with the input mean (and variance). In contrast to the other demographic parameters where estimated values vary by +/- 30 % among the four species, estimated clonal growth rates varied by ~ 600 % (see Table S1 in Hulber et al. 2016). Taking the median value of the estimated range for clonal growth rates allowed taking into account this variation. Lastly, the sensitivity of the growth rate to fecundity is low (see main answer).

- The info presented in the matrices implies no seedbank exists for any of these species. Can the authors clearly show evidence that this is the case? The existence of a seedbank can drastically affect the population dynamics of plant species...

See above. Addition of a seedbank in *Dianthus* and *Primula* only slightly changed the species' population dynamics.

- If I were to do this analyses, I would do it with species for which a lot of demographic information (in situ) has been collected and published for many years and locations (and the ideal repository for that already exists - www.compadre-db.org), rather than pull from thin air demographic information. I would also perhaps include an expert in demography to help with the stochastic modeling of vital rates and potentially with the dispersal kernels - the latter might have been done correctly, but with the limited info at hand the authors have provided, I cannot state further.

While the compadre-database is really very useful, a recent meta-analysis has shown that there are only 136 published studies which have monitored demography across environmental gradients, and only 28 of them where focusing on climatic gradients, which are of particular importance in the context of this paper (Ehrlen et al. 2016). Put it another way, demographic information for many years and many (environmentally sorted) sites is still available for very few species globally. Here, we wanted to focus on the species which are particularly valuable from a conservation perspective (because they are regional endemics), for which as much information as possible is available, and which grow in an area that we are familiar with and where comparable studies that do not account for evolutionary processes have already been conducted (the Alps). The four species finally selected represent an optimal intersection of these criteria.

We think that while the demographic parameters used for our species are still relatively rough estimates, they nevertheless provide a reasonable demographic model integrating the main characteristics of our species both in absolute and relative terms. The kernels have not been calculated anew, but have been taken as such from Dullinger et al. (2012). A detailed description of how these kernels have been constructed is provided in this paper (and its supplementary material).

There are many other points I have made in the PDF. Most of them have to do with my inability to objectively evaluate aspects of the model because not enough details have been provided (e.g. dispersal information, soil information, detail son True Skill Statistic), or because figure legends are not fully coherent.

In the new version of the manuscript, we improved the description of the methods and of the main model assumptions in the introduction (around L86-117)

PS: Code (from Caswell 2001) to estimate mean life expectancy, where U is the matrix without reproductive rates:

```
colSums(solve(diag(4)-U))[1]
```

To fully answer reviewer 1, we provide a complete description of species demography in the supplementary materials (*Life cycle* section of the Supp. Mat.).

Reviewer #2 (Remarks to the Author):

Review of NCOMMS-16-18804 "Slow eco-evolutionary response of alpine plants to climate warming"
Jorge Soberón, University of Kansas

This paper describes the application of a modeling exercise that combines niche modeling with dispersal and natural selection on the characters defining the niche. It is based on well-studied alpine plants, and presents interesting results, both in terms

of fundamental ecology, and of relevance to adaptation to climate change. The results are novel (but see below) and general, and certainly new for the plants and the ecosystem in question. Although many antecedents to this paper can be found in the theoretical literature, the results in the context of niche modeling and climate change are most likely unappreciated. Moreover, the described methodology should become a welcome addition to the toolbox of students of ranges of distribution under climate change. I believe the paper should be published.

Having said that, I still have a few comments. The main one is related to the lack of mention to the theoretical antecedents of the work. There are quite a few papers highly relevant to this work that are not mentioned, and they should, since the authors are essentially exploring range dynamics under climate change, a topic with a rich literature. Their discussion (mainly lines 96-119) will gain clarity and force if some previous results were actually incorporated: Holt and coworkers have discussed these problems since long ago (Holt & Gomulkiewicz, 1997a; Holt & Gomulkiewicz, 1997b) showing that, under very general conditions, subpopulations outside the limits of the niche will go extinct before natural selection can rescue them. This result does not depend on any spatial structure, but it can probably be useful to understand the results in the manuscript.

Other factors hindering the expansion of areas of distribution (Kawecki, 2008; Sexton et al., 2009) include gene flow from populations adapted to conditions in the core of niche space to populations outside or marginal to the niche. This was first mentioned by Haldane (Haldane, 1956) half a century ago and is very relevant to the work under review. Recent theoretical results (García-Ramos & Kirkpatrick, 1997; Kirkpatrick & Barton, 1997) confirm those early hypothesis. In the simulation in the MS this effect is almost certainly taking place but it is not mentioned. I suggest that using the results will add value to the discussion.

We would like to thank the reviewer for these comments. We are aware of this previous literature. For concision, we chose to focus mostly on the literature aiming at providing species range projections. We think that our study mostly add an (eco-)evolutionary dimension to the niche modeling approach but is not new regarding the underlying evolutionary model as the reviewer points out. We however agree that our evolutionary approach relies on previous developments that ought to be mentioned.

In addition to the reviewer suggestion for the above literature on the limit to range expansion by migration and maladaptation, we would like to point to recent theoretical articles showing that when the (classical) assumption of a fixed genetic variance is relaxed in the quantitative

genetics framework, migration does not limit species range anymore (Barton 2001). A recent article suggests that genetic drift in peripheral populations can generate sharp limits of species range (Polechova and Barton 2015).

Our approach using individual based simulations is related to these above studies because it considers the changes in genetic variance and both demographic and genetic stochasticity. Our results in the absence of climate suggests that the range of all species slowly increases with time (not shown), which is consistent with the above results. It is also why we prevented range expansion during the burn-in phase: without constraining migration for many generations, the species range would inflate unrealistically.

Nevertheless, many features of now classical eco-evo quantitative eco-evo modeling apply to our model as we now point along the main text. In particular, we point out that most extinctions occur in peripheral populations, which is consistent with the results from all studies pointed out by reviewer 2.

In the new version of the manuscript we integrated references to the previous work on adaptation and range expansion (L71-72; 454-455)

Finally, in a rather obscure paper (and in Spanish... <http://www.misclaneamatematica.org/Misc49/4906.pdf>), Soberon & Miller (Soberón & Miller, 2009) report results of a simulation of natural selection on a two dimensional niche, on a grid of populations, with and without spatial covariance on the environmental factors. They show that the spatial covariance of environmental values (and therefore of selective pressures) is determinant of the details of the evolutionary dynamics of the niche, and of total population size. Again, I suspect that the simulation described in the MS being reviewed is also affected by spatial covariance but the authors do not include this point in their discussion. It may be interesting simply to present variograms of the environmental values as climate changes, and discuss whether climate is moving towards higher or lower spatial covariance (higher covariance facilitates evolution, lower covariance slows it down).

There are spatial covariations between the environmental variables we chose in our study. However, for simplicity, we assumed no genetic correlation between the corresponding quantitative traits. In other words, spatial covariations have no effect on the evolutionary response of population because each trait responds to the new environment independently. However, we agree that, in real populations, genetic correlations are likely to affect the adaptation dynamics and hence demography. We added this point in the new version of the manuscript L254-263.

Many of the above results can be used to compare, contrast and explain the results of the paper's simulation.

There are several little details:

1) It is rather puzzling to see the paper justified (L41-43) by reference to documents of bureaucratic bodies. Maybe cite chapter and verse of the reports of IPBES that are relevant to a paper about the evolution of range-sizes? I rather would like to see reference to some of the conceptual papers by Holt, Kirkpatrick, Kawecki or any of those, but this really is up to the authors.

2) L55-57. The factors mentioned here are precisely the factors that the papers quoted above, and not referred to, analyze!

We have added reference to relevant literature in this section (now L71-72)

3) Line 68. Substitute "real" by "realistic"?

Done

4) L436-437. Downscaling WorldClim data to 100 meters is probably extremely unadvisable, since at 100 meters weather is really becoming microclimate, influenced by topography (even microtopography), habitat, wind, soil... none of which is included in the original WorldClim database. Downscaling without recalculating using such microclimate-relevant parameters is essentially a numerical exercise, with no real value. Please justify this step.

Working on a relatively fine-scale when modelling spatio-temporal dynamics of herb and grass species is essential to represent dispersal process which mostly operate on small distances. While we agree with the referee that downscaling of WORLDCLIM to 100 m will not be able to accurately capture the microclimate of this 100 m sites, the alternative would have been to assume the same climate for all 100 x 100 m cells within 1 WORLDCLIM site, which measures about 1 km² in the Alps. At steep slopes 1 km² can capture a gradient of several °C mean annual temperature (which decreases by about 0.6°C per 100 elevational meters). Hence, even if inaccurate, our downscaling, that accounts for the elevational temperature gradient, certainly improves the accuracy of the climatic variables as opposed to the assumption that the climate is constant across 1 km in high mountain terrain.

5) L449-450. Seems to me that the interval are of 14 years, not 17... Please check or clarify.

We changed to 14 years

6) L480-482. What algorithm/s was/were used by Biomod2? With what settings? Presence-absence or presence only? An ensemble? Of what? There are no details whatsoever in this very important part of the modeling process. The fact that the authors use

(L483) "probability of occurrence of a species" suggest they were using some variety of logistic regression with presence-absence data. If this is not the case, then correct this as well.

We used true presence absence data from botanical surveys. The methods used in Biomod2 are now shortly detailed in the manuscript. We also made a more in-depth description of the procedure in the Supplementary methods and made the script used available on GitHub.

7) There is no description of the settings of the simulator Nemo. This is a highly sophisticated program, with many parameters. I suggest that the details of the simulation, including the external text files with parameters are added in the supplementary materials.

Details of the simulations are provided in the method section and in the corresponding reference to the simulator (Guillaume and Rougemont 2006)

We think that the best option is to upload Nemo parameter files on a data repository. We can provide these files to the editor and reviewers if they are requested.

García-Ramos, G. & Kirkpatrick, M. (1997) Models of adaptation and geneflow in peripheral populations. *Evolution*, 51, 21-28.
Haldane, J. B. S. (1956) The Relation between Density Regulation and Natural Selection. *Proceedings of the Royal Society of London. Series B, Biological Sciences*, 145, 306-308.
Holt, R. D. & Gomulkiewicz, R. (1997a) The evolution of species niches: a populations dynamic perspective. *Case Studies in Mathematical Modelling: Ecology, Physiology and Cell Biology* (ed. by H.G. Othmer & F.R. Adler & M.A. Lewis & J. Dillon), pp 25-50. Prentice-Hall, Englewood Cliffs, NJ.
Holt, R. D. & Gomulkiewicz, R. (1997b) How Does Immigration Influence Local Adaptation? A Reexamination of a Familiar Paradigm. *The American Naturalist*, 149, 563-572.
Kawecki, T. J. (2008) Adaptation to marginal habitats. *Annual Review of Ecology, Evolution and Systematics*, 39, 321-342.
Kirkpatrick, M. & Barton, N. H. (1997) Evolution of a species range. *The American Naturalist*, 150, 1-23.
Sexton, J. P., McIntyre, P. J., Angert, A. L. & Rice, K. J. (2009) Evolution and ecology of species range limits.
Soberón, J. & Miller, C. P. (2009) Evolución de los nichos ecológicos. *Miscelánea Matemática*, 49, 83-99.

Reviewer #3 (Remarks to the Author):

This paper presents a modelling exercise using a novel expansion of niche based species distribution models, DEEM. Curiously enough this acronym is not spelled out in the text, but these models incorporate short-term evolutionary response, thus enabling examination of eco-evolutionary dynamics. The models are used to predict responses of four long-lived alpine plants to projected climate change over the coming 135 years (until the year 2150). The main conclusions are that these long-lived plants will have difficulties to adapt to climate change, eventually becoming more 'maladapted', and that their populations therefore will decline. The response is however delayed due to the plant's adult life-span. This will contribute to an extinction debt.

The main value of the study lies in the approach, which can be useful to predict species response to climate change, and in the sense that the predicted results can inspire more detailed studies of the eco-evolutionary dynamics of species (although for such studies short-lived plants would be more suitable).

Focusing on the specific results, on the other hand, the conclusion that long-lived organisms has a slow evolutionary response (in an absolute time-scale) is of course well-known, and will hardly surprise anyone.

We agree that the general eco-evolutionary processes that we describe are not new and relatively straightforward. We think that the benefit of our approach is to estimate how they act on a spatial scale to predict species range in response to an environmental forcing such as climate change. Our revised models, however, now allow us to highlight the importance of the feedback between demography and evolutionary dynamics of our species, which may cause rapid extinction depending on species specific parameter values.

As far as I can judge the modeling is technically accurate. My only technical question is whether the lack of fine-scale resolution may limit the applicability of the models. The climatic variables were downscaled to just 250 m resolution, meaning that microclimatic variation, which are now increasingly recognized as very important, not the least in alpine environments, are neglected.

We agree that fine-scale climatic variation can be important for species distribution in alpine environments. However, capturing this variation in models like this faces two important limitations. First, the accurate representation of the relevant microclimate is difficult, and, second, there are important computational limitations when running a complex-evolutionary

model like this at a fine grain over a relevant extent (beyond one single mountain chain), The 250 m resolution chosen here is a compromise which already includes heavy computational demand (e.g., simulations of a high-occupancy grid with *Festuca* may require up to 24GB of RAM and last about 11h for 5000 years in the burn-in). At this scale, some relevant fine-scale variation is certainly neglected. On the other hand, a recent comparative analysis has demonstrated that stepping down from 100 m to 1 m grain size does not improve accuracy of alpine plant distribution models essentially – it seems that others than climatic factors (e.g. soil attributes) are more important at this resolution (Pradervand et al. 2014). We hence think that the grain size selected here is not optimal but tolerable for the purpose of a study that deals with eco-evolutionary response to a changing climate.

However, as always in modelling the critical issue is the assumptions. I had a few remarks on these, which may necessitate some clarification.

The basic model (a static niche model) was based on one soil and two climatic variables. Many alpine plants are also strongly affected by grazing and/or management in high-altitude meadows. Now, I am not familiar with the chosen plant species, but would appreciate a comment on other factors, not included in the models, that may be important for these plants.

All four species modelled here have their current center of distribution at elevations beyond those mainly used for livestock pasturing, i.e. in the alpine, not the subalpine belt. *Campanula pulla* is moreover a specialist of rocky calcareous snowbeds poor in forage. Main grazers within the ranges of our study plants are wild herbivores like chamois or ibex, which occur in rather low densities, however. We hence think that neglecting grazing pressure in models of these species is justifiable.

In addition, we mention that the approach we present is applicable to situations where other gradients are important.

A key assumption is that populations are locally adapted. Thus, the environment changes, whereas the plants (due to their long adult life span) genetically do not. Over time, the plants become maladapted. But what is the basis for assuming that the plants were adapted in the first place? There are several reasons why they may not have been perfectly locally adapted. Firstly, previous climate has also changed; climate change occurs all the time. And if the generation time is long, these plants may over a couple of generations have experienced different climate regimes (e.g. the 'little ice age' between the 15th and 19th centuries). What is the rationale for assuming that the plants today are locally adapted to current climate?

We agree that populations may not be perfectly adapted to their local environment. However, we think that it is a reasonable assumption for those species, and the way we proceeded allowed to soften this effect. First, several studies have found evidence for local adaptation in Alpine plant populations. Even though this does not show that populations are adapted in the sense of absolute fitness, there is still the information that local individuals perform better than foreigners do, which is consistent with our local adaptation assumption. In addition, and in accordance with reviewer 1 comments, the actual lifespan of our species is long relative to the rate of climate change. The lifespan of our focal species is however not large relative to past climatic fluctuations. This suggests that populations have undergone several generations of selection under an environmental series with a roughly constant mean. Lastly, our approach softens the effect of the initial perfect adaptation by allowing 5000 generations of burn-in where migration and mutation generated phenotypic variation around the local trait optima in local populations. Thus, prior to climate change, the populations in our simulations contain variation at least representative of the diversity of habitats occupied. The variation locally maintained at mutation-selection-migration equilibrium actually causes a reduction of mean population fitness.

Nevertheless, we tested the effect of local adaptation with a set of simulations where all individuals have the same genotype corresponding to the mean niche value. While this alternative scenario assumes the opposite extreme of no local adaptation, and absence of evolution, we would find difficult to justify imposing an arbitrary level of maladaptation without explicitly modeling past-climate series. Unfortunately, this is outside the scope of this work, although it might be relevant for the exercise we set forth.

Secondly, as clonal plants have an erratic and sporadic recruitment, there might be very weak selection on the recruitment and juvenile phase. Thus, the surviving adults may just have been lucky, rather than equipped with the 'best' locally adapted genotypes.

In our simulations, we take into account the wave of recruitment effect by modeling the inter-individual competition for recruitment. When a cohort of adults dies, it releases competition, in turn allowing a wave of seedling recruitment. Even though seedlings that survive to the adult stage are on average adapted, the stochasticity included in the individual based simulation framework further allow some maladapted seedlings to be recruited by chance.

So, what would happen if the assumption of an initial local adaptation is relaxed? Does anything then at all happen related to eco-evolutionary dynamics?

See above

In the model, selection operates only on juveniles. This may seem reasonable, but the risk is that two other aspects of selection are neglected. Selection on the adult phase may operate if fecundity varies in response to climate change. Given

the rare and sporadic recruitment episodes in many clonal plants, fecundity may actually be important. In addition, for clonal plants one should not downgrade the potential importance of within-genet selection (which may occur in clonal organisms).

We performed exploratory simulations with selection on adult fecundity only (see main answer).

The competition process described above take into account the within-genet competition as seedlings produced by cloning are in competition with other seedlings and with adults, regardless of their genotype.

References:

- Barton, N. 2001. Adaptation at the edge of a species' range. *Special Publication-British Ecological Society* **14**:365-392.
- Chambers, J. C., J. A. Macmahon, and R. W. Brown. 1990. Alpine seedling establishment - The Influence Of Disturbance Type. *Ecology* **71**:1323-1341.
- Elmendorf, S. C., and K. A. Moore. 2008. Use of community-composition data to predict the fecundity and abundance of species. *Conservation Biology* **22**:1523-1532.
- Ehrlen, J., Morris, W.F., von Euler, T. & Dahlgren, J.P. (2016) Advancing environmentally explicit structured population models of plants. *Journal of Ecology* 104: 292-305.
- Forbis, T. A. 2003. Seedling demography in an alpine ecosystem. *American Journal of Botany* **90**:1197-1206.
- Gonzalo-Turpin, H., and L. Hazard. 2009. Local adaptation occurs along altitudinal gradient despite the existence of gene flow in the alpine plant species *Festuca eskia*. *Journal of Ecology* **97**:742-751.
- Guillaume, F., and J. Rougemont. 2006. Nemo: an evolutionary and population genetics programming framework. *Bioinformatics* **22**:2556-2557.
- Hülber, K., Wessely, J., Gattringer, A., Moser, D., Kuttner, M., Essl, F., Leitner, M., Winkler, M., Ertl, S., Willner, W., Kleinbauer, I., Sauberer, N., Mang, T., Zimmerman, N.E., Dullinger, S. 2016: Uncertainty in predicting range dynamics of endemic alpine plants under climate warming. *Global Change Biology* **22**: 2608-2619.
- Keller, R. & Vittoz, P. 2015: Clonal growth and demography of a hemicryptophyte alpine plant: *Leontopodium alpinum* Cassini. *Alpine Botany* **125**: 31-40.
- Kuss, P., M. Rees, H. H. Ægisdóttir, S. P. Ellner, and J. Stöcklin. 2008. Evolutionary demography of long-lived monocarpic perennials: a time-lagged integral projection model. *Journal of Ecology* **96**:821-832.
- Pauli, H., M. Gottfried, S. Dullinger, O. Abdaladze, M. Akhalkatsi, J. L. B. Alonso, G. Coldea, J. Dick, B. Erschbamer, and R. F. Calzado. 2012. Recent plant diversity changes on Europe's mountain summits. *Science* **336**:353-355.

- Polechová, J., and N. H. Barton. 2015. Limits to adaptation along environmental gradients. *Proceedings of the National Academy of Sciences* **112**:6401-6406.
- Pradervand, J.-N., Dubuis, A., Pellissier, L., Guisan, A. & Randin, C. (2014) Very high resolution environmental predictors in species distribution models: moving beyond topography? *Progress in Physical Geography* 1: 79-96
- Schwienbacher, E., Marcante S., Erschbamer B. (2010) Alpine species seed longevity in the soil in relation to seed size and shape – A 5-year burial experiment in the Central Alps. *Flora* 205: 19-25.
- Steinger, T., Körner, C. & Schmid, B. 1996: Long-term persistence in a changing climate: DNA analysis suggests very old ages of clones of alpine *Carex curvula*. *Oecologia* 105: 94-99.
- Thompson, K., Band S.R., Hodgson J.G. (1993) Seed size and shape predict persistence in soil. *Functional Ecology* 7: 236-241.
- Weppeler, T., P. Stoll, and J. Stöcklin. 2006. The relative importance of sexual and clonal reproduction for population growth in the long-lived alpine plant *Geum reptans*. *Journal of Ecology* **94**:869-879.
- Willner W, Berg C, Heiselmayer P (2012) Austrian vegetation database. In: *Vegetation Databases for the 21st Century* (eds Dengler J, Oldeland J, Jansen F, Chytrý M, Ewald J, Finckh M, Glöckler F, Lopez-Gonzalez G, Peet RK, Schaminée JHJ), pp. 333.

Reviewers' Comments:

Reviewer #1 (Remarks to the Author):

The work by Cotto and collaborators "Slow eco-evolutionary response of alpine plant to climate warming" builds climatic projections for the spatially explicit viability of four plant species from the Alps of, as the authors state, high longevity, and various modes of life history before various climatic scenarios. The authors find that the persistence of adults in non-suitable habitats allows the species' populations to persist beyond what models not incorporating demographic information predict. This supposedly would result in a buffer against local and eventually total extinction by all four species, though at various rates, which all would eventually result in demographic failure.

The authors have implemented some (but overall not all) the comments and main analytical suggestions that I made wrt the critical assumptions made with their demographic approach and study species - See the author's responses to previous review.

The fact that the authors have not obtained more reliable information on a critical vital rate for this type of life cycle, or opted to use one of the many* study species for which rich demographic information exist in open-access format, instead, renders the interpretation of their results rather limited to an expert demographer.

Growth rates extracted from the matrix can't possibly be "potential maxima" as the authors state, since arguably most of the underlying vital rates come from natural settings, so the species' dynamic is the result of realised niche dynamics. The point of unusually high rates remains thus a point of limitation in this study.

*The authors discuss species choice based on conservation concerns - plenty of the species in the suggested COMPADRE database are also of conservation concern, take place in similar areas in Europe, and have much more reliable data than the one used here. The fact that Ehrlén's J Ecol 2016 review highlights about 20 herbaceous plants for which rich data that would be useful in this question exist is not a limitation - the authors have 20 species to choose from! I don't think their choice to stick to their species under the big assumptions they have been forced to make is warranted for Nature Communications.

I am really sorry I have not been able to be more positive in this particular case. Very interesting scientific approach, but scarcity of data and assumptions on life cycle remains the Achilles heels of this paper, imo.

Reviewer #2 (Remarks to the Author):

None.

Reviewer #3 (Remarks to the Author):

I have no further comments.

REVIEWERS' COMMENTS:

Reviewer #1 (Remarks to the Author):

The work by Cotto and collaborators "Slow eco-evolutionary response of alpine plant to climate warming" builds climatic projections for the spatially explicit viability of four plant species from the Alps of, as the authors state, high longevity, and various modes of life history before various climatic scenarios. The authors find that the persistence of adults in non-suitable habitats allows the species' populations to persist beyond what models not incorporating demographic information predict. This supposedly would result in a buffer against local and eventually total extinction by all four species, though at various rates, which all would eventually result in demographic failure.

In addition to demography, our simulations integrate evolutionary dynamics. One of the main outcome of our study is to link species demographic properties to evolutionary dynamics: the persistence of adult delays the decrease in the species range and slows down the evolutionary adaptation to the new climate.

The authors have implemented some (but overall not all) the comments and main analytical suggestions that I made wrt the critical assumptions made with their demographic approach and study species - See the author's responses to previous review.

The fact that the authors have not obtained more reliable information on a critical vital rate for this type of life cycle, or opted to use one of the many* study species for which rich demographic information exist in open-access format, instead, renders the interpretation of their results rather limited to an expert demographer.

In the revision of the manuscript, we did our best to answer the reviewer concerns about species demography. To do so, we refined the calculation of the demographic rates and performed a sensitivity analysis to the adult survival rate, a key parameter of the study. Even though our demographic model is not perfectly accurate, we think that it reasonably captures the demographic properties of the species we studied. We thus expect our results to be relatively general regarding the life cycle of herbaceous Alpine plant species. In particular, our sensitivity analysis reinforced the conclusions from the first version of the manuscript on the role of adult persistence.

Importantly, our framework requires more than demographic data. Knowledge of the seed dispersal kernels and the geographical information necessary to the niche models (especially a mapping of presence-absence points) are equally important. While COMPADRE provides a high quality database for demographic data, sufficient information on dispersal and species distribution is not necessarily available for the species in this database. Good quality data on both demography, dispersal, and geographical range, as well as climatic data and projections, were available for the species we chose.

In the revised version of the manuscript we acknowledge the existence of the COMPADRE database and suggest that our framework would be especially suited to species in this database for which distribution data are available at an appropriately fine spatial resolution over a sufficiently large extent to cover the ecological niche. We hope that our study will stimulate further collection and analytical combination of such data.

Growth rates extracted from the matrix can't possibly be "potential maxima" as the authors state, since arguably most of the underlying vital rates come from natural settings, so the species' dynamic is the result of realised niche dynamics. The point of unusually high rates remains thus a point of limitation in this study.

We based our analysis on the impact of the environment and density dependence of seedling survival. As explained in the manuscript, this assumption is supported by previous studies. In addition, a preliminary sensitivity analysis showed that the growth rate of the population (as given by the leading Eigenvalue of the transition matrix) is especially sensitive to seedling survival. The results of this preliminary analysis are further confirmed by the fact that selection on adult fecundity leads to very different results, where there is almost no effect of the environment.

As a basis for the seedling survival rate, we used the juvenile survival rate as given in Hülber et al. 2016 GCB. In this last study, this rate is estimated from "well suited environmental condition and without density dependence" (see their table S1). In addition, most other vital rates in this table S1 refer to maxima observed individually in samples of different populations, i.e. they represent maxima that did not, in most cases, occur jointly. Thus, the population growth rates estimated from the combinations of these optima will overestimate what is observable in most real populations. In our simulations, we included maladaptation and density dependence as dynamic variables, so that the actual growth rates in the simulations are always far below the rates estimated from raw values. This is the reason why we consider the growth rates extracted

from the demographic transition matrix as “potential maxima”. Obviously, the growth rates are not that high in the simulations once all processes have been taken into account.

*The authors discuss species choice based on conservation concerns - plenty of the species in the suggested COMPADRE database are also of conservation concern, take place in similar areas in Europe, and have much more reliable data than the one used here. The fact that Ehrlén's JEcology 2016 review highlights about 20 herbaceous plants for which rich data that would be useful in this question exist is not a limitation - the authors have 20 species to choose from! I don't think their choice to stick to their species under the big assumptions they have been forced to make is warranted for Nature Communications.

See answer above.

I am really sorry I have not been able to be more positive in this particular case. Very interesting scientific approach, but scarcity of data and assumptions on life cycle remains the Achilles heels of this paper, imo.

Reviewer #2 (Remarks to the Author):

None.

Reviewer #3 (Remarks to the Author):

I have no further comments.